# “When You’re Smiling”: How Posed Facial Expressions Affect Visual Recognition of Emotions

**DOI:** 10.3390/brainsci13040668

**Published:** 2023-04-16

**Authors:** Francesca Benuzzi, Daniela Ballotta, Claudia Casadio, Vanessa Zanelli, Carlo Adolfo Porro, Paolo Frigio Nichelli, Fausta Lui

**Affiliations:** Department of Biomedical, Metabolic and Neural Sciences, University of Modena and Reggio Emilia, 41125 Modena, Italy

**Keywords:** emotion recognition, facial expressions, emotions, empathy, fMRI

## Abstract

Facial imitation occurs automatically during the perception of an emotional facial expression, and preventing it may interfere with the accuracy of emotion recognition. In the present fMRI study, we evaluated the effect of posing a facial expression on the recognition of ambiguous facial expressions. Since facial activity is affected by various factors, such as empathic aptitudes, the Interpersonal Reactivity Index (IRI) questionnaire was administered and scores were correlated with brain activity. Twenty-six healthy female subjects took part in the experiment. The volunteers were asked to pose a facial expression (happy, disgusted, neutral), then to watch an ambiguous emotional face, finally to indicate whether the emotion perceived was happiness or disgust. As stimuli, blends of happy and disgusted faces were used. Behavioral results showed that posing an emotional face increased the percentage of congruence with the perceived emotion. When participants posed a facial expression and perceived a non-congruent emotion, a neural network comprising bilateral anterior insula was activated. Brain activity was also correlated with empathic traits, particularly with empathic concern, fantasy and personal distress. Our findings support the idea that facial mimicry plays a crucial role in identifying emotions, and that empathic emotional abilities can modulate the brain circuits involved in this process.

## 1. Introduction

The importance of facial expressions: In 1872, Darwin published *The Expression of the Emotions in Man and Animals* [1]. This pioneering work highlighted the idea that emotions are universal and discrete entities expressed particularly through the face. 

In the last two centuries, psychologists and neuroscientists confirmed the central role of facial expression in emotional processing. 

Consistent evidence has supported Darwin’s idea of discrete emotional categories characterized by several facial movements that vary to some degree around a typical set of movements. This approach assumes that there is a core facial configuration—the prototype—that can be used to detect the emotional state of an individual. Variations in expressions are ascribed to non-emotional processes such as display rules, emotion-regulation strategies (e.g., suppressing the expression) or culture-specific effects [2,3,4,5,6,7].

The Facial Feedback Hypothesis: The expression and experience of emotion seem to be strictly linked. Once again, the idea was introduced by Darwin, who noted that the experience of an emotion seemed to be intensified when the emotion was freely expressed, and softened when suppressed [1]. This framework is now known as the facial feedback hypothesis or, more recently, embodied emotion [8,9,10]. The latter term refers to the idea that the observed facial expression triggers a simulation of a state in the motor, somatosensory, affective and reward systems, representing the meaning of the expression to the perceiver. The central hypothesis of the embodied emotion theory is that the sensorimotor system is the main contributor to the visual recognition of facial expressions and to other socially relevant tasks, such as action recognition [11,12,13,14], and social interactions including empathy [15,16,17]. The visual perception of an emotional facial expression activates a somatosensory and motor pattern that largely overlap with that subserving the production of the same facial expression [18,19]. This reactivation of a facial expression via a sensorimotor simulation is thought to occur by means of facial mimicry [18,20,21]. Finally, the sensorimotor simulation of facial expressions activates the associated emotional system of the observer, who, experiencing the same emotional state of the other person, uses this information to recognize the facial expression seen [18]. Several studies showed that facial feedback can modulate present emotions and can induce new emotions [22,23,24]. This effect was found for happiness and anger [25,26,27,28], fear and sadness [29] and surprise and disgust [30].

However, a recent meta-analysis [31] found that the effect of facial feedback measured through emotional self-reporting was significant but small, and modulated by several variables. For instance, they showed that facial movements have larger effects on initiating than modulating the emotional status of the subjects, and that presenting emotional audio or imagined scenarios had a greater effect than pictures, video-clips and stories. On the other hand, this revision did not find any effect of the following variables: the discrete versus dimensional measures of emotional experience, the awareness of facial feedback manipulation and of being video-recorded and the gender of the volunteers. 

Facial mimicry: Irrespective of the fact that facial feedback can modulate the subjective experience of emotions, facial mimicry, i.e., the tendency to unconsciously and unintentionally imitate the facial expressions of others, is well documented [32,33,34,35]. In most cases, this phenomenon is undetectable to the eye, but it can be evaluated using electromyography (EMG), with, for instance, a similar muscle response to that observed being detected within one second of the facial expression being presented [20,36,37]. For example, an enhanced EMG activity of the *zygomaticus major* muscle (the muscle that causes the corners of the mouth to rise during smiling) is observed when a person sees a happy face and an increase in the activity of the *corrugator supercilii* muscles (the muscles that moves the eyebrow down and inward toward the nose and inner eye to frown) in response to an angry expression [32]. 

Mimicry has been considered a “social glue” [38] because it can generate a feeling of similarity which in turn promotes prosocial behavior [39]. In a recent fMRI study, spontaneous facial mimicry activated the reward neural system, and the magnitude of this effect was positively correlated with trait empathy, thus, emphasizing the “reward value of the act of mimicking”. Other studies confirmed the relationship between facial mimicry and levels of individual empathy, but also between facial mimicry and the susceptibility to emotional contagion [40,41,42,43,44].

Several studies have demonstrated that altering spontaneous facial mimicry affects the recognition of emotions in others [41,45,46,47,48,49], modulates the visual working memory representations of facial expressions [50], activates several cortical areas and modulates the activity of emotional regions such as the insula, anterior cingulate and amygdala [51,52]. 

Although there is general agreement on the notion that visual and sensorimotor cues provide congruent information in decoding a specific and unique facial expression, there is no agreement on how this simulation process is neurally implemented and to what extent facial mimicry is crucial for emotion recognition.

In the present fMRI study, we evaluated the effect of sensorimotor information on the perception of emotion by asking subjects to pose a facial expression while viewing ambiguous emotional faces. The volunteers were asked to pose a facial expression (happy, disgusted or neutral), watch an image representing a real face expressing an emotion and indicate whether the emotion perceived was happiness or disgust. Ambiguous faces were obtained as a blend of disgusted and happy faces. We also evaluated the effect of empathy on this neural network, as the ability to react emotionally to the emotional expressions is one aspect of empathy. Moreover, within the framework of the facial feedback hypothesis, facial mimicry may be a key for empathy, as the facial muscles function as a feedback system for a person’s own experience of emotion.

Facial mimicry is influenced by various motivational and contextual factors such as individual traits and, specifically, empathic tendencies [18,34,53]. Previous studies have found that emotional empathy, as opposed to cognitive empathy, predicts the extent of spontaneous facial mimicry in response to facial expressions [42,44,54,55]. Consequently, we administered the Interpersonal Reactivity Index (IRI) questionnaire [56], a self-administered questionnaire used to assess emotional and cognitive aspects of empathy independently. Given previous research on facial mimicry [53], we hypothesized that trait empathy could modulate the neural circuit that subserves the emotional processing of facial expression.

## 2. Materials and Methods

### 2.1. Participants

Twenty-six right-handed young women (mean age: 23.7; range: 18–39 years; mean school age: 14; range: 13–18) took part in the fMRI study. Handedness was assessed using the Edinburgh Inventory [57] and the participants had no history of neurological or psychiatric diseases. Since previous studies suggested a gender difference in empathy [58,59], only female volunteers were included. The experimental protocol was approved by the local Ethics Committee and all subjects gave their written informed consent to take part in the study.

### 2.2. Stimuli

The stimuli were ambiguous emotional faces, blends of happy and disgusted faces selected from the Ekman series [60]. Two identities from the series were used. The following blends were used: 50% happy, 50% disgusted; 55% happy, 45% disgusted; 60% happy, 40% disgusted. Neutral faces were also employed for a total of 4 stimuli per identity. Stylized emotional faces (emoticons—happy, H, disgusted, D, neutral, N) were used as a cue for the posed emotional expression to be assumed at the beginning of the trial. 

### 2.3. Procedures

An event-related fMRI paradigm was used. Each subject performed 4 sessions comprising 27 trials each, for a total of 108 trials. Each trial lasted 14 s and began with a 500 ms change of the background color from black to blue (visual warning cue). Then, the volunteers were asked to pose an emotional facial expression (H, D, N) according to a stylized face (emoticon) that appeared on the screen (1.5 s), to keep posing the expression while watching a real human face with an ambiguous facial expression (1 s; total duration of the pose = 2.5 s) and, after a 9 s interval, to indicate whether the emotion expressed by the ambiguous face was happiness (h) or disgust (d; 2 s; see Figure 1). Two passive rest blocks lasting 15 and 24 s were included at the beginning and at the end of each session, respectively. Stimulus presentation was counterbalanced across the four sessions. Participants gave their response by pressing one of two buttons on a response pad that we provided at their right hand. Behavioral responses were collected during the scanning sessions by means of custom-made software developed in Visual Basic 6 (http://digilander.libero.it/marco_serafini/stimoli_video/, accessed on 10 September 2014). The same software was used to present stimuli via the ESys functional MRI System (http://www.invivocorp.com, accessed on 20 September 2014) remote display.

Subjects were not video-recorded during the fMRI scanning sections because of safety issues in the MRI environment. For this reason, we cannot provide an illustration of the emotion posed by the participants. However, the accuracy of the mimicry was controlled via facial electromyography (EMG) recorded using an MRI-compatible EMG recording system (Micromed, Mogliano Veneto, Italy) consisting of three bipolar and one reference electrode. The electrodes (Sintered Detection Cup Electrodes) were positioned in pairs over the *corrugator supercilii*, *levator labii* and *zygomaticus major* muscles [61,62] on the left side of the face. The reference electrode was attached to the left shoulder. Before the electrodes were attached, they were filled with electrode paste and the skin was cleaned with alcohol. The electrode impedance was reduced to 10 kΩ. The digitized EMG signals were transmitted via an optic fiber cable from the high-input impedance amplifier to a computer located outside the scanner room. During fMRI acquisition, a TTL signal was sent every TR (repetition time) via a BNC (Bayonet Neill Concelma), a trigger cable from the MRI console to the EMG computer allowing the synchronization between the acquisition of the functional volumes and the EMG data. The correction of the gradient-echo pulse artifacts was performed offline using the average artefact subtraction (AAS) method [63] implemented in the Brain Quick System Plus software (Micromed, Mogliano Veneto, Italy). The EMG data of each participant were qualitatively analyzed by an expert neurophysiopathology technician to ascertain the congruency between the required facial pose and the muscle activity. 

At the end of the scanning session, the volunteers completed the Interpersonal Reactivity Index (IRI) [64,65]. The IRI is a self-report rating index designed to measure personal empathy defined as the “reactions of one individual to the observed experiences of another” [66]. It contains twenty-eight items and four subscales (perspective taking, PT; fantasy, FS; empathic concern, EC; personal distress, PD).

### 2.4. Behavioral Data Analyses

An arc-sine transformation was run on the percentages of disgust and happiness responses to obtain a normal distribution. A 3 (pose: D, F and N) × 2 (response: h and d) within subjects ANOVA was conducted and a Newman–Keuls post-hoc test was used. Spearman’s rank-order correlation coefficient was calculated to check the correlations between responses and IRI scores.

### 2.5. fMRI Data Acquisition and Analyses

Functional data were acquired using a Philips Achieva system at 3T and a gradient-echo echo-planar sequence from 30 axial contiguous slices (TR = 2000 ms; in-plane matrix = 64 × 64; voxel size = 3.75 mm × 3.75 mm × 4 mm) over four 651 s sessions per participant. A high-resolution T1-weighted anatomical image was acquired for each participant to allow anatomical localization. The volume consisted of 170 sagittal slices (TR = 9.9 ms; TE = 4.6 ms; in plane matrix = 256 × 256; voxel size = 1 mm × 1 mm × 1 mm).

fMRI analysis was performed using Matlab version R2013a (The MathWorks Inc., Natick, Mass) and the standard SPM12 (Statistical Parametric Mapping, Wellcome Department of Imaging Neuroscience, London, UK) approach. Functional volumes of each participant were slice-time corrected, realigned to the first volume acquired, normalized to the MNI (Montreal Neurologic Institute) template implemented in SPM12 and smoothed with an 8 mm × 8 mm × 8 mm FWHM Gaussian kernel. Due to excessive movement during the scanning, the last two sessions were excluded in three subjects, and the last session was excluded in five subjects.

Functional data of each participant were first analyzed individually and then fed into second-level random effect analyses. By means of the general linear model implemented in SPM12, a 3 × 2 factorial design analysis was performed. The first factor represented the pose (happy, H, disgusted, D, and neutral, N), whereas the second factor was the subject’s response, i.e., happiness (h) or disgust (d). Each condition was modeled by convolving the stimulus onset (ambiguous emotional face) and each motor response with the standard hemodynamic response function (HRF) to create regressors of interest. Motion parameters obtained from the realignment were used as additional regressors of no interest. According to the aim of the study, the linear contrasts of “incongruent (posing disgust and perceived happiness, Dh, and posing happiness and perceiving disgust, Hd) versus neutral (posing neutral and perceived happiness, Nh, and posing neutral and perceived disgust, Nd)” and “congruent (posing disgust and perceived disgust, Dd, and posing happiness and perceiving happiness, Hh) versus neutral” were used to study the effect of pose; the resulting contrast images were entered in the random effects group analyses.

Finally, regression analyses were used to explore which brain regions showed a correlation with individual empathic personality traits assessed using the post-scanning questionnaires.

A family-wise error (FWE) correction was used for the “Incongruent vs. Neutral” and “Congruent vs. Neutral” contrasts, whereas a double statistical threshold (voxel-wise *p* < 0.001 and spatial extent) was adopted for regression analyses to correct for multiple comparisons; the latter combined significance of α < 0.05 was assessed using the 3dClustSim AFNI routine, using the “-acf” option (https://afni.nimh.nih.gov/pub/dist/doc/program_help/3dClustSim.html, accessed on 28 November 2018).

For all analyses, the Matthew Brett correction (mni2tal: http://www.mrc-cbu.cam.ac.uk/Imaging/mnispace.html, accessed on 28 November 2018) was applied to the SPM-MNI coordinates to obtain the coordinates in Talairach space [67].

## 3. Results

The qualitative analysis of the EMG data confirmed that the participants actually posed the required emotional facial expressions for each trial.

### 3.1. Behavioral Data

The ANOVA revealed a significant effects of the factor response (happiness responses were more frequent than disgust ones; F = 70.5; *p* < 0.001; df = 1; *n* = 26; power = 1) and of the interaction between the response and pose (F = 6.6; *p* = 0.001; df = 2; *n* = 26; power = 0.9; Figure 2). Post-hoc tests showed significant differences between posing disgust and responding with disgust vs. posing disgust and responding with happiness (*p* = 0.02); and posing happiness and responding happiness vs. posing happiness and responding disgust (*p* = 0.04).

No significant correlation between behavioral data and the IRI scores (subtest and total scores) was found. 

### 3.2. fMRI Data

#### 3.2.1. Effect of Posed Emotions on Perceived Emotions

The contrast incongruent vs. neutral evaluated areas of significant signal changes for the incongruent conditions (i.e., when subjects posed an expression and perceived a different one), as contrasted with neutral conditions (no pose). Increased activations were detected in a wide range of cortical and subcortical regions, including the bilateral anterior insula (AI), right pre- and postcentral gyri and cerebellum (Table 1 and Figure 3: the activation shown here survived FWE correction). A similar pattern of activation was found for the contrast of congruent vs. neutral, but at a lower, non-significant level.

#### 3.2.2. Correlations with Empathy Subscales

Empathic Concern Subscale

A significant positive correlation was found between the activation of the left precuneus and superior parietal lobule and the empathic concern scores when the subjects were posing and perceiving happiness as compared to posing happiness and perceiving disgust (Hh vs. Hd; Table 2 and Figure 4).

Fantasy subscale

A positive correlation with the fantasy score was found with the right IFG/AI and caudate nucleus and left temporal cortex (inferior, middle and superior temporal gyri) and supramarginal gyrus when posing disgust and perceiving happiness as compared to posing and perceiving happiness (Dh vs. Hh; Table 3 and Figure 5).

Personal distress subscale

The comparison between perceived happiness and perceived disgust, irrespective of the pose (h vs. d), showed a positive correlation between personal distress scores and right superior parietal lobule, angular gyrus, cuneus and precuneus, left pre- and postcentral gyrus, IFG/AI and inferior parietal lobule, bilateral occipital cortex, fusiform gyrus and cerebellum. (Table 4 and Figure 6).

In addition, we found a significant positive correlation with the right lingual gyrus and bilateral cerebellum when posing with a neutral expression and perceiving happiness compared to posing with a neutral expression and perceiving disgust (Nh vs. Nd) (Table 5; Figure 7).

No significant correlation was found for the PT subscale.

## 4. Discussion

In the present event-related fMRI study, we evaluated whether posing facial expressions can support automatic emotion recognition. We asked participants to pose a facial expression (happy, disgusted or neutral) while judging the emotion of visually presented ambiguous faces. Our behavioral results showed that posing an emotion shifts the visual perception of ambiguous expressions towards that same emotion: posing a disgusted face increased the proportion of disgust responses, whereas posing a happy face increased the proportion of happiness responses. 

Several studies demonstrated that humans usually voluntarily imitate or unconsciously match the nonverbal behaviors of others: these phenomena are called mimicry. Mimicry has been demonstrated to happen in response to facial expressions [32], body movements [68] or even pupil dilations [69]. Facial mimicry is often imperceptible and can be detected only using specialized and sensitive methods that can measure contractions of the facial muscles accurately. Using electromyography (EMG), muscle reactions matching observed facial or bodily expressions can be detected within a second of exposure. For instance, observing positive emotional facial expressions can lead to heightened muscle activity in the *zygomatic major*, while negative expressions can activate the *corrugator supercilii* within 500 milliseconds of the presentation of the stimulus [32,61,70].

Facial expression recognition is supported by visual expertise that is partially responsible for the capability to extract information from faces [71]. In addition, the sensorimotor simulation employed by reproducing the motor movements of the observed facial expressions can facilitate the recognition of the emotional expression. This motor activity, which is typically unconscious, is thought to elicit partial activation in the neural circuits involved in experiencing the corresponding emotion. The simulator can then implicitly deduce the internal state of the person displaying the expression. In support of this hypothesis, few studies have evaluated the impact of facial movement on the ability to recognize emotions, particularly from facial expressions [18]. For instance, Ponari et al. [47] developed a series of experiments in which they manipulated the participants’ ability to move either the upper or lower half of their face. Their results indicated that the “lower” manipulation specifically impaired recognition of happiness and disgust, while the “upper” manipulation hindered recognition of anger, and both manipulations affected the recognition of fear. Wood et al. [18] prevented participants from producing facial movements by applying a gel facemask, and asked subjects to distinguish between target facial expressions and similar-looking distractors. They found that the participants’ ability to recognize emotions was impaired, indicating that the sensory and motor processes linked to expression imitation contribute to the visual perceptual processing of facial expressions. More recently, Borgomaneri et al. [41] evaluated whether inhibiting mimicry affects the recognition of happiness conveyed through facial or body expressions. Their results showed that blocking mimicry on the lower face affected the recognition of happy facial and body expressions, while the recognition of neutral and fearful expressions was unaffected by the experimental manipulation. 

Taken together, these findings support the role of facial mimicry in emotion recognition, and suggest that facial mimicry may reflect a comprehensive sensorimotor simulation of others’ emotions, rather than just a muscle-specific replication of an observed motor expression.

Our functional results showed that when participants posed a facial expression and perceived a non-congruent emotion (for instance, posing disgust and perceived happiness, or posing happiness and perceiving disgust) a neural network comprising the bilateral anterior insula, motor cortex, cerebellum and superior temporal gyri was activated. This pattern of activation was also present in the case of congruency between the pose and perception, but at a lower, non-significant level. 

A similar pattern of fMRI signal changes was detected by Braadbaart et al. [72]. In this study, participants were instructed to either imitate (match) or perform mismatched facial movements in response to blends of facial emotional expressions. Their results showed greater neural activity during the execution of mismatched actions as compared to imitation, especially in the insula bilaterally. 

The involvement of the anterior insula could be ascribed to its known role in detecting conflict between action and response, as described by Ullsperger et al. [73]. Our results supported their hypothesis that this involvement is not surprising, given the insula’s role in both the visual perception of facial expressions and the self-expression of the same emotion [74,75]. It is reasonable to expect that the insula’s role in error monitoring would result in stronger responses to facial stimuli that do not match the actions being performed, leading to greater sensitivity to incongruency between the pose and response. The activation of the motor cortex and the cerebellum could be ascribed to the motor component of the imitation task. 

Our functional results also showed that some brain regions exhibit correlations with the individual empathic disposition, as tested using the different IRI subscales.

A previous study demonstrated that empathic disposition could modulate facial feedback in response to emotional expressions. Some studies have shown a correlation between levels of individual empathy or susceptibility to emotional contagion and facial mimicry [44,53,54], although the strength of this relationship remains uncertain. 

Williams et al. [76] proposed that imitation is the link between facial mimicry and empathic abilities. The authors argued that while facial mimicry may utilize a primary sensorimotor representation [77,78,79], imitation may require a secondary representation of the intention and the motor plan for that action [80]. Facial imitation may, therefore, involve mechanisms similar to those involved in empathy; indeed, empathy deals with communicating emotions and needs a secondary representation of those emotions. In turn, this representation enables emotion understanding [81]. This hypothesis is linked to the simulation theory of empathy, which suggests that empathizers use their neural systems to imitate actions “offline” in order to imagine and understand the experiences of others [82]. Moreover, this hypothesis is in line with the perception–action model of empathy [83], which suggests that empathy depends on the perception–action coupling mechanisms necessary for imitation. Evidence for a strict link between empathic traits and imitation abilities comes from autism studies that suggested that, in these patients, an impairment in empathy and imitation co-occurs [84]. Moreover, a limited range of facial expressions is considered to have diagnostic value for autism [85]. In support of this view, Williams et al. [76] developed an imitation paradigm in which participants were asked to imitate expressions of faces representing a blend of emotions. The accuracy of imitation was rated by two experimenters and correlated with the empathy quotient (EQ) score. The EQ is a 60-item self-report questionnaire that measures individual differences in empathic ability [81]. The results showed that participants who scored a higher EQ exhibited superior facial imitation skills, especially when imitating more complex stimuli.

The link between empathic abilities and facial mimicry could be the facilitation effect of facial imitation in understanding others’ emotions. As already reported above, Borgomaneri et al. [41] recently showed that blocking mimicry on the lower face impaired the recognition of happy facial and body expressions. Furthermore, this impairment was correlated with empathic traits. Specifically, the index of the drop in accuracy of emotion recognition was significantly correlated with the empathic concern (EC) IRI subscale score, that is, individuals with lower levels of emotional empathy were significantly impaired when mimicry was blocked, whereas those with higher levels of emotional empathy showed little or no interference.

Our functional results showed a significant correlation between EC scores and the activation of the left precuneus and superior parietal lobule when congruent facial mimicry facilitated a happiness response (i.e., when the subjects were posing happiness and perceiving happiness, as opposed to posing happiness and perceiving disgust). These brain regions are the functional core of the Default Mode Network (DMN), reflecting self-referential processes that are active during the resting state. 

The DMN plays a role in evaluating survival-relevant information from the body and the world [86]. This includes subsuming the other’s point of view, desires, beliefs and intentions, as well as remembering the past and planning for the future [87]. These functions are all self-referential in nature. When engaged in a cognitive processing effort, there is a reduction in activity in the DMN [86,88]. This reduction can be interpreted as an adaptive mechanism to reduce self-referential activity in the brain and improve focus on the task. Failure to do so may result in interference from internal emotional states during task performance, as observed in patients with depression [86]. More recently, precuneus activity was found to correlate with the score of subjective happiness (SHS). In particular, trainees of mindfulness meditation [89], which reportedly heightens SHS scores [90] and increases the gray matter volume in the precuneus [91], showed reduced resting state precuneus activity when compared with participants who did not follow the training [92]. In patients with depression, who exhibit low SHS scores [93] and a reduced gray matter volume in the precuneus [94], resting-state blood flow in this region was increased during depressive episodes [95,96] and was decreased after a significant improvement in their mental health [97]. 

The sensorimotor simulation of others’ emotions in subjects with higher emotional empathic abilities is correlated with the activity of the precuneus and could reflect their higher capacity to emotionally empathize with other people’s state of mind and to “resonate” with their positive feelings. 

We also found a positive correlation with the fantasy score, in particular with the activity of the right IFG/AI and caudate nucleus, left temporal cortex (inferior, middle and superior temporal gyri) and supramarginal gyrus when posing a disgusted expression and perceiving happiness compared to posing and perceiving happiness. These areas, particularly the IFG/AI and caudate nucleus, are consistently activated during disgust processing [75] and could reflect the ability of people with a high capacity for imagination to feel disgust as if they were actually experiencing it.

Finally, we found a positive correlation between the activity of several posterior areas, the anterior insula and sensorimotor areas and the personal distress score when the participants perceived happiness compared to when they perceived disgust. PD measures the tendency to react with a negative emotional self-focused response to the apprehension or comprehension of another’s emotional state or condition. These activations may be considered in line with other studies, e.g., the study by Llamas-Alonso et al. [98], who detected higher levels of activity in the occipital regions with regards to happy faces than angry faces during a pro-saccade task; and a study by Loi et al. [99], who, in an EEG study, found that the view of happy, but not sad, faces increased the excitability in face M1 bilaterally. Furthermore, recent research [100] related levels of subjective well-being with increased gray matter volumes of the anterior insula. In addition, Luo et al. [101] investigated the correlation of DMN activity with subjective happiness and greater connectivity was found in several cortical areas, including the inferior parietal lobule in people with a high level of subjective unhappiness. These results were interpreted as suggesting that this abnormal activity may indicate excessive negative self-reflection. However, in our protocol, activity in these areas was correlated with the PD score. We speculate that in subjects with higher aptitude to personal distress, these regions are even more related to the perception of positive emotions (happiness) compared to the general population.

## 5. Conclusions

To the best of our knowledge, this is the first study to investigate emotional face processing under ambiguous conditions and personality traits, namely, empathic aptitudes. Our behavioral results revealed that posing an emotion shifts the visual perception of ambiguous expressions towards that same emotion. Our functional results showed that the brain activity underlying this processing is modulated by individual emotional empathic disposition, as tested using the different IRI subscales. In particular, we found a significant correlation between empathic concern (EC) scores and the activation of regions that represent the functional core of the Default Mode Network, reflecting self-referential processes that are active during the resting state. We also found a positive correlation between the fantasy (FS) score and the activity of areas known to be correlated with disgust processing [75], which could reflect the ability of people with a high capacity for imagination to feel disgust as if they were actually experiencing it. Finally, we found a positive correlation between the personal distress (PD) score and the activity of several posterior areas, the anterior insula and sensorimotor areas. PD measures the tendency to react with a negative emotional self-focused response to the apprehension or comprehension of another’s emotional state or condition. According to the literature, we speculate that in subjects with a higher aptitude to personal distress, these regions are even more related to the perception of positive emotions (happiness) compared to the general population.

Based on our results, we suggest that under ambiguous conditions, the prevalence of bottom–up sensory stimulation or top–down motor priming is determined by individual characteristics. 

## Figures and Tables

**Figure 1 brainsci-13-00668-f001:**
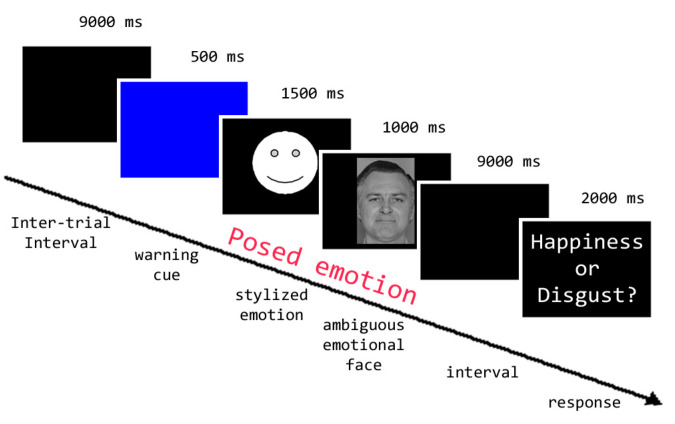
Experimental protocol for the fMRI session.

**Figure 2 brainsci-13-00668-f002:**
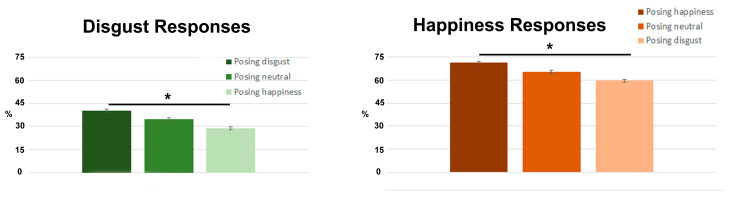
Behavioral results. Left: % of disgust responses for participants posing disgust/neutral/happiness expression. Right: % of happiness responses for participants in happiness/neutral/disgust expression. * = *p* < 0.005, bars = standard errors.

**Figure 3 brainsci-13-00668-f003:**
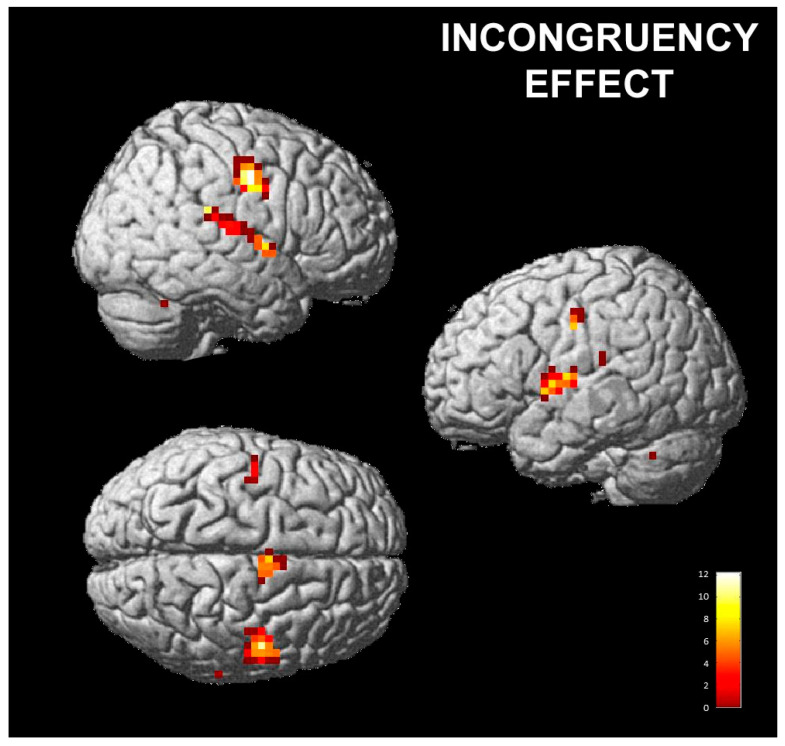
Incongruent pose vs. neutral pose one-sample t-test; *p* < 0.05, FWE corrected. Functional results are shown on the SPM12 template; color bars represent T-values.

**Figure 4 brainsci-13-00668-f004:**
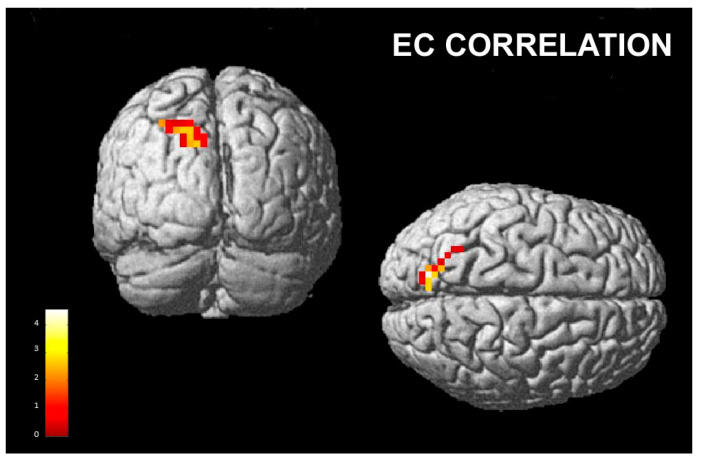
EC correlation for the contrast “posing happiness and perceiving happiness” vs. “posing happiness and perceiving disgust”; cluster size threshold k > 30, corrected at α < 0.05. Same overlay procedure as in Figure 3.

**Figure 5 brainsci-13-00668-f005:**
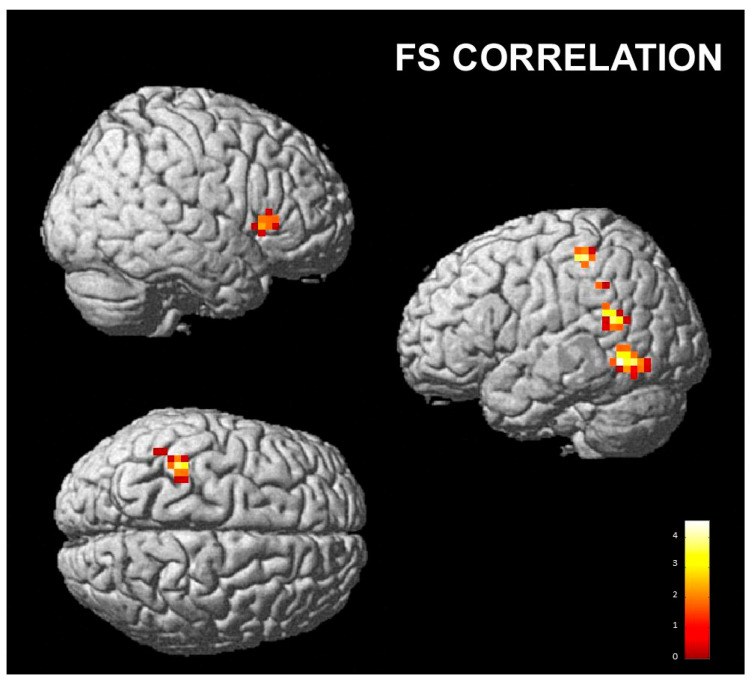
FS correlation with the contrast “posing disgust and perceiving happiness” vs. “posing and perceiving happiness”; cluster size threshold k > 31, corrected at α < 0.05. Same overlay procedure as in Figure 3.

**Figure 6 brainsci-13-00668-f006:**
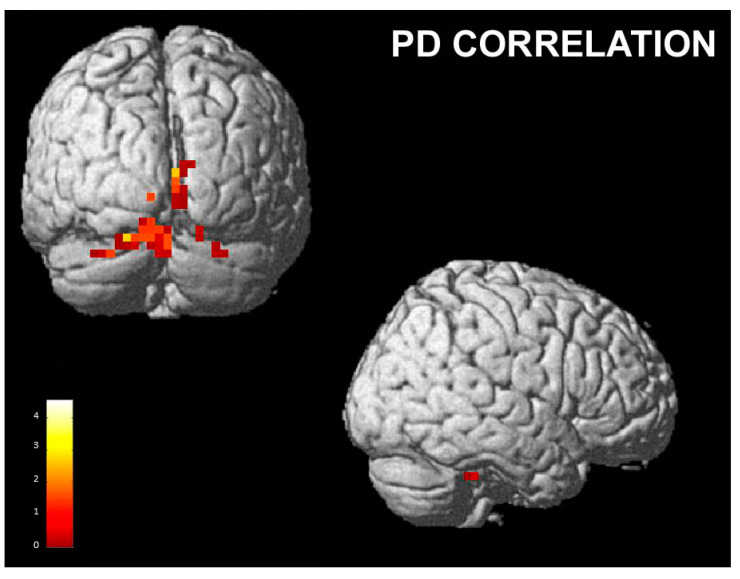
PD correlation with the contrast “perceived happiness” vs. “perceived disgust”; cluster size threshold k > 31, corrected at α < 0.05. Same overlay procedure as in Figure 3.

**Figure 7 brainsci-13-00668-f007:**
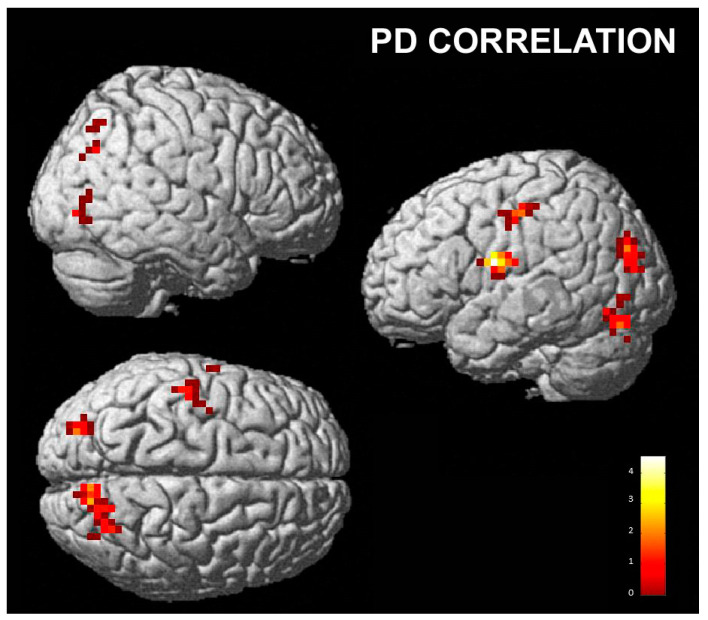
PD correlation with the contrast “posing neutral and perceiving happiness” vs. “posing neutral and perceiving disgust”; cluster size threshold k > 27, corrected at α < 0.05. Same overlay procedure as in Figure 3.

**Table 1 brainsci-13-00668-t001:** Peak coordinates of functional activation related to incongruent conditions.

	BA	Side	Cluster	Voxel Level	MNI Coordinates			Talairach Coordinates		
Brain Areas			K	T	x	y	z	x	y	z
*Incongruent vs. Neutral*										
Precentral gyrus, postcentral gyrus	4, 6	r	53	12.19	46	−8	38	46	−6	35
Cerebellum		l	49	8.76	−18	−60	−22	−18	−59	−16
				6.18	−38	−56	−34	−38	−56	−34
Cerebellum		r	28	8.25	22	−60	26	22	−57	27
				7.59	30	−60	−30	30	−59	−22
Postcentral gyrus, inferior parietal lobule, superior temporal gyrus	40, 41, 42	r	21	8.14	62	−20	14	61	−19	14
				7.31	62	−28	18	61	−26	18
				6.08	54	−32	22	53	−30	22
Thalamus		r	6	7.62	14	−8	2	14	−8	2
Operculum, precentral gyrus, superior temporal gyrus, insula	6, 13, 22, 43, 44	l	39	7.4	−50	−12	10	−50	−11	10
				7.16	−50	−4	6	−50	−4	6
				6.69	−58	4	6	−57	4	5
Superior temporal gyrus, operculum, insula, precentral gyrus	6, 13, 22	r	27	7.3	62	0	2	61	0	2
				6.88	46	−8	10	46	−7	10
				6.15	50	4	−2	50	4	−2
Cerebellum		r	28	7.29	6	−4	62	6	−1	57
				6.3	2	4	66	2	7	60
Precentral gyrus	6	l	13	7.18	−42	−12	42	−42	−10	39
Inferior parietal lobule	40	l	2	6.16	−58	−28	22	−57	−26	22
Cerebellum			1	5.94	2	−36	−2	2	−35	0

Areas of significant changes in fMRI signal for the contrast of “incongruent pose” vs. “neutral pose”; BA = Brodmann area; l = left; r = right; FWE corrected.

**Table 2 brainsci-13-00668-t002:** Peak coordinates of positive correlation with the empathic concern subscale.

	BA	Side	Cluster	Voxel Level	MNI Coordinates			Talairach Coordinates		
Brain Areas			K	T	x	y	z	x	y	z
*Hh vs. Hd*										
Precuneus, superior parietal lobule	7	l	34	4.53	−10	−76	38	−10	−72	39
				4.38	−14	−72	50	−14	−67	49
				4.14	−22	−64	38	−22	−60	38

Areas of significant correlation with EC scores for the contrast “posing happiness and perceiving happiness” vs. “posing happiness and perceiving disgust”; BA = Brodmann area; l = left; r = right; cluster size threshold k > 30, corrected at α < 0.05.

**Table 3 brainsci-13-00668-t003:** Peak coordinates of positive correlation with the Fantasy-Empathy subscale.

	BA	Side	Cluster	Voxel Level	MNI Coordinates			Talairach Coordinates		
Brain Areas			K	T	x	y	z	x	y	z
*Dh vs. Hh*										
Anterior insula, inferior frontal gyrus	45, 47	r	16	4.9	42	20	2	42	19	1
				3.79	50	28	6	50	27	4
Postcentral gyrus	3	l	24	4.81	−38	−36	54	−38	−32	51
Inferior temporal gyrus	37	l	35	4.69	−50	−64	−10	−50	−62	−5
Superior temporal gyrus		l	36	4.48	−46	−52	18	−46	−50	19
				4.39	−42	−44	34	−42	−41	33
				4.1	−54	−48	22	−53	−45	23
Caudate nucleus		r	21	4.24	14	0	10	14	0	9
				3.77	22	−12	18	22	−11	17

Areas of significant correlation with FS scores for the contrast “posing disgust and perceiving happiness” vs. “posing and perceiving happiness”; BA = Brodmann area; l = left; r = right; cluster size threshold k > 31, corrected at α < 0.05.

**Table 4 brainsci-13-00668-t004:** Peak coordinates of positive correlation with the personal distress subscale.

	BA	Side	Cluster	Voxel Level	MNI Coordinates			Talairach Coordinates		
Brain Areas			K	T	x	y	z	x	y	z
*h vs. d*										
Cerebellum		r	30	5.76	30	−40	−30	30	−40	−23
Cuneus, superior parietal lobule, angular gyrus	7, 39	r	104	5.44	10	−76	38	10	−72	39
				4.81	18	−68	46	18	−64	46
				4.81	26	−64	46	26	−60	45
Pre- and post-central gyrus		l	36	4.9	−50	−20	50	−50	−17	47
				4.26	−34	−12	42	−34	−10	39
Cuneus, lingual gyrus, PCC, cerebellum	18, 19	r	131	4.86	6	−72	14	6	−69	16
				4.09	18	−64	−6	18	−62	−2
				3.95	18	−52	−18	18	−51	−13
Inferior frontal gyrus, anterior insula, postcentral gyrus	43	l	69	4.82	−38	8	14	−38	8	12
				4.78	−54	−4	18	−53	−3	17
				4.52	−46	−16	18	−46	−15	17
Middle and superior occipital cortex		l	39	4.6	−26	−80	22	−26	−76	24
				4.34	−30	−88	14	−30	−85	17
Middle occipital gyrus, fusiform gyrus	19	l	33	4.14	−46	−76	−2	−46	−74	−2
				4.12	−38	−72	−18	−38	−71	−12
				4.12	−30	−80	−14	−30	−78	−8

Areas of significant correlation with PD scores for the contrast “perceived happiness” vs. “perceived disgust”; BA = Brodmann area; l = left; r = right; cluster size threshold k > 31, corrected at α < 0.05.

**Table 5 brainsci-13-00668-t005:** Peak coordinates of positive correlation with the Personal Distress subscale.

	BA	Side	Cluster	Voxel Level	MNI Coordinates			Talairach Coordinates		
Brain Areas			K	T	x	y	z	x	y	z
*Nh vs. Nd*										
Lingual gyrus, cerebellum	18	r	144	4.89	22	−72	−6	22	−70	−2
				4.29	30	−68	−30	30	−67	−22
				4.12	22	−76	−18	22	−74	−11
Cerebellum		r	52	4.65	22	−40	−30	22	−40	−23
				4.03	18	−52	−18	18	−51	−13
				3.74	22	−44	−14	22	−43	−10
Lingual gyrus, cerebellum	18	l	56	4.31	−2	−76	−22	−2	−75	−15
				4.06	−14	−76	−26	−14	−75	−18
				3.87	−30	−72	−30	−30	−71	−22

Areas of significant correlation with PD scores for the contrast “posing neutral and perceiving happiness” vs. “posing neutral and perceiving disgust”; BA = Brodmann area; l = left; r = right; cluster size threshold k > 27, corrected at α < 0.05.

## Data Availability

The data presented in this study are available on request from the corresponding author. The data are not publicly available due to the privacy policy.

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
