# Peer review of "“When You’re Smiling”: How Posed Facial Expressions Affect Visual Recognition of Emotions"

_brainsci, 2023, doi:10.3390/brainsci13040668_

Round 1
Reviewer 1 Report
Dear Author,
I have carefully read your manuscript titled "When you’re smiling”: how posed facial expressions affect visual recognition of emotions," and I must say that I find your work to be very interesting and informative. You present a clear research question and provide a comprehensive analysis of the results that contributes significantly to the field of emotion recognition.
I particularly appreciate the study design, which was well thought out and appropriate for the research question. The methods were described clearly and in sufficient detail, making it easy to follow the procedures used. Additionally, the data analysis was rigorous, and the results were presented in an organized and understandable manner.
The discussion section was particularly strong, as it provided a thoughtful interpretation of the findings and highlighted the implications of your work for both theory and practice. However, one area of concern is the relatively small sample size, which may limit the generalizability of the results. I suggest considering this limitation in your discussion section and providing ideas for future research that may address this issue.
Overall, I find your manuscript to be well-written, well-organized, and well-researched. The findings presented in this paper have important implications for the field of emotion recognition and may have practical applications in fields such as psychology and marketing.

Author Response
According to the request, we have added two additional keywords to the original three (Page 1, line 24).
With respect to References, we have used the standard Brain Sciences ‘References guidelines: references have been numbered in order of appearance in the text and listed individually at the end of the manuscript.
We have checked the accuracy of the Reference both in text and in the References list.
According to the Reviwer's request, we have changed the Figure legends (please see Page 6 lines 229-234; Page 8, lines 263-264; Page 9 lines 276-277; Page 10 lines 286-287; Page 11, 302-303).
In the Methodology section, according to the reviewer’s request, we have changed Figure 2 and changed the Conclusions sessions (Please see pages 14-15 lines 464-477). However, we could not catch the point with respect to the experimental method and abstract: which part does the Review find difficult to understand and need clarification?
We would like to thanks Reviewer 1 for his/her suggestions on the two References. However, can the Reviewer indicate the point in the text where these citations should be added?

Reviewer 2 Report
The authors present an interesting topic facial expressions affect visual recognition of emotions. Introducerea studierii emotiilor faciale cu ajutorul jMRI fata de metodele traditionale app software de imagistica cu algoritmi de Inteligenta Artificiala aduce un plus in aceasta analiza. From a medical point of view, the article is complete, but from a technical point of view, some clarifications are needed:
for the statistical analysis, the SPM 12 application developed by a UK institute was used. I ask the authors to present in the article how the statistical analysis is carried out, which are the algorithms used for the voxel cluster, for example: 1) parametric statistics based on Random Field Theory); 2) nonparametric statistics based on permutation/randomization analyses; and 3) nonparametric statistics based on Threshold Free Cluster Enhancement.
- the SPM analysis method mentioned in the article was also used by other authors in articles reported in the specialized literature for similar medical cases. What is the novelty that the authors propose from the point of view of the proposed method?
Author Response
Thanks to the Reviewer for his/her positive comments. According to his/her suggestions, we have added an abbreviation list to the manuscript (please see Pages 15-16). We have also improved the quality of Figure 2 (we apologize for inadvertently uploading an imperfect version of this figure the first time).
Regarding the illustration of the posed emotions, subjects were not video-recorded during the fMRI scanning sections, because of safety issues in the MRI environment. For this reason, we can not provide an illustration of the emotion posed by the participants. However, the accuracy of the mimicry was controlled via the EMG recordings (Please see Methods pages 3 and 4: lines: 144-164).
Reviewer 3 Report
I think the authors have submitted an interesting manuscript which is written in a good English. The manuscript is easy to read and follow. I didn't find any logical errors in the presentation. In the manuscript, the authors use a lot of abbreviations. This is why, a list of abbreviations section at the end of manuscript would be helpful for readers (mdpi template provides an opportunity for this). The quality of Figure 2 is very low. Please improve it. Illustration of posed emotions would be also very helpful for readers.
Author Response
We would like to thank the reviewer for his/her positive comments.
The focus of the present study was not to suggest novel analysis methods. Therefore, we have analysed fMRI data using SPM (Statistical Parametric Mapping): SPM12 uses a classical parametric approach that considers statistic image as lattice representation of a continuous random field and uses random field theory to solve the multiple comparison problem. We have used the typical, suggested, standard approach, as stated in the methodological section. We have now added a brief note that underlies our classical, standard fMRI data approach (Please see page 4 lines 181-182).
Round 2
Reviewer 2 Report
I have no more comments.
Author Response
Thanks for his/her positive comment.